# A High Sensitivity Temperature Sensing Probe Based on Microfiber Fabry-Perot Interference

**DOI:** 10.3390/s19081819

**Published:** 2019-04-16

**Authors:** Zhoubing Li, Yue Zhang, Chunqiao Ren, Zhengqi Sui, Jin Li

**Affiliations:** 1College of Information Science and Engineering, Northeastern University, Shenyang 110819, China; lizhoubing50@163.com (Z.L.); 20174113@stu.neu.edu.cn (Y.Z.); Rencq12138@163.com (C.R.); 15942415413@163.com (Z.S.); 2State Key Laboratory of Synthetical Automation for Process Industries, Northeastern University, Shenyang 110819, China

**Keywords:** fiber sensors, temperature sensors, Fabry-Perot interferometer, microfiber, PDMS, integrated optics

## Abstract

In this paper, a miniature Fabry-Perot temperature probe was designed by using polydimethylsiloxane (PDMS) to encapsulate a microfiber in one cut of hollow core fiber (HCF). The microfiber tip and a common single mode fiber (SMF) end were used as the two reflectors of the Fabry-Perot interferometer. The temperature sensing performance was experimentally demonstrated with a sensitivity of 11.86 nm/°C and an excellent linear fitting in the range of 43–50 °C. This high sensitivity depends on the large thermal-expansion coefficient of PDMS. This temperature sensor can operate no higher than 200 °C limiting by the physicochemical properties of PDMS. The low cost, fast fabrication process, compact structure and outstanding resolution of less than 10^−4^ °C enable it being as a promising candidate for exploring the temperature monitor or controller with ultra-high sensitivity and precision.

## 1. Introduction

As a typical physical parameter, the temperature must be carefully controlled and monitored in many fields, such as clinical medicine, biochemical reactions, industrial production, aviation safety and so on [1,2,3]. In recent years, optical fiber temperature sensors have aroused widespread research interest, because of their unique advantages compared with electrical ones, such as remote monitoring capability, high sensitivity, anti-electromagnetic interference properties, and intrinsic safety [4,5]. By combining the resonance enhancement effect of the optical coupling technique, multi-modes interference, optical evanescent field, optical time domain reflecting and optical ring-down technology produced by different special optical fiber structures, various optical fiber temperature sensors were realized [6,7,8,9]. Multi-modes interference is carried out by splicing together different kinds of fibers to excite the modes’ interference. The splicing joints are fragile and the length for each section must be carefully controlled during the fabrication process; Optical evanescent fields can be obtained around micro/nanofibers with diameters comparable to the wavelength of the incident light. Although micro/nanofibers offer excellent performance, the sensor probes based on them are difficult to fabricate because of their thin diameter and environmentally sensitive properties. The optical time domain reflection technique was used to sense temperature and strain based on Raman or Brillouin scattering [10]. The sensitivity of the temperature sensor based on optical ring-down technology only can be increased by extending the fiber length.

In addition to the basic sensing mechanism, the sensing performance was further improved by means of temperature sensitive materials [11]. Many materials, such as polymers and metal oxides, have been reported to be elaborated by surface or inner coating, and used to encapsulate the whole fiber structure [12,13,14]. In addition to the temperature dependence, the effect of humidity, strain and other related parameters on the sensing performance must be determined and eliminated. At present, the most common commercial optical fiber temperature sensor is the fiber Bragg grating (FBG) having good repeatability and stable sensing characteristics [15]. It can be prepared by ultraviolet exposure or nano-etching technology to meet the working requirements of different temperature ranges [16]. However, high sensitivity or precision is difficult to obtain for FBG temperature sensors, which seriously hinders their commercial application [17].

The optical fiber temperature sensors based on multi-wavelength interference mainly include Mach–Zehnder and Fabry-Perot interferometers. The former typically perform as transmission structures, which separate and transmit an independent signal light and reference light by using different special optical fibers or structures, such as micro/nano fibers, photonic crystal fibers (PCFs), dislocation fusion fibers and multi-core fibers. A corresponding sensitivity of up to 6.5 nm/°C was observed [18]. However, the structures of the Mach-Zehnder interferometers are complex due to their dual-optical-paths system [19,20,21,22]. To simplify the structures, the two optical paths can be revealed in single fiber, named the in-line Mach-Zehnder interferometer, such as C-typed PCFs [23], side-hole PCFs [24], D-shaped-hole fibers [25] and muti-core fibers [26]. These compact structures were precisely machined using femtosecond lasers, focused ion beams and chemical vapor deposition, and display excellent stability and sensing performance. However, these are hard to manufacture in batches due to the high cost and technical requirements. In addition to the above complex optical fiber structures, single polymer optical fibers have been demonstrated with a temperature sensitivity of ~10^−3^ °C [27], where the temperature performance were revealed by the transmission power and the effect of relative and twist have been experimentally obtained [28,29]. Furthermore, their packaging size is hard to reduce further depending on the bending loss of the optical fiber [30], which will seriously limit their application in a narrow space; the latter ones are carried out as reflective structures, where the temperature sensitive cavity was constructed at the end of the optical fiber by laser or ion beam processing, chemical etching or film forming and special fiber splicing technologies [31,32,33,34,35,36,37]. Among them, femtosecond laser processing can machine a refractive index turning point with good repeatability in the optical fiber, which was used as a Fabry-Perot cavity and can work at high temperatures up to 1000 °C [31]; focused ion beams can etch an air cavity at the tip of an optical fiber, based which a Fabry-Perot temperature sensor with a sensitivity of −654 pm/°C has been experimentally demonstrated [32]. However, the expensive and complex preparation processing, as well as the high technique requirements for engineers have become huge obstacles for commercial production [33]. The Fabry-Perot interferometer probe can be obtained conveniently and quickly by chemical etching or film forming technology [34], however, the fabrication repeatability is low, and the structural parameters are difficult to control accurately [35]. By using the special hollow-core photonic bandgap fiber (HC-PBF) or PCF, the temperature working range and sensitivity of cascaded splicing fiber based Fabry-Perot interferometer has been experimentally verified as high as 1200 °C and 17 nm/°C, respectively, but their structures are relatively fragile [36,37].

Compared with conventional temperature sensors, the proposed Fabry-Perot interferometer temperature sensor costs less and is easier and faster to prepare. This compact Fabry-Perot temperature probe was proposed by encapsulating a microfiber and a single mode fiber (SMF) tip in a hollow core fiber (HCF), between which temperature sensitive polydimethylsiloxane (PDMS) was filled and cured. The microfiber was prepared by the one-step heating-stretching technique from a normal SMF. The microfiber and SMF can be easily aligned due to the comparable inner diameter of HCFs. The high transparency and low refractive index of PDMS causes little impact on the incident light. Furthermore, a sensitivity of higher than 11 nm/°C has been experimentally demonstrated due to its high thermal expansion coefficient. This temperature sensor will be a promising candidate for monitoring temperature fluctuations in small spaces due to its high sensitivity and tiny scale (200 μm in diameter and <5 mm in length).

## 2. Materials and Methods

To fabricate the Fabry-Perot interferometer, a cut of transparent HCF was prepared firstly, as shown in Figure 1. The coating layer of a HCF (TSP150200, inner diameter: ~150 μm, outer diameter: ~200 μm, coating layer: polyimide, Polymicro Technologies, Inc., Phoenix, AZ, USA) was removed by a Bunsen burner (Dragon 200, fuel: butane, max-temperature 1300 °C, Rocker Scientific Co., Ltd., New Taipei, Taiwan), as shown in inset (a) of Figure 1. The cavity length of Fabry-Perot interferometer can be observed through its transparent wall. Both the microfiber and SMF can be inserted and aligned easily due to their small diameter difference. The microfiber was obtained from the SMF (Coating removed diameter: 125 μm, SMF-28, Corning Inc., Corning, NY, USA) using the scanning flame heating-stretching technique (inset (b) of Figure 1). Where, the diameter and length of microfiber were precisely controlled by optimizing the fabrication process of a fiber melting-drawing system (IPCS-5000-ST, Idealphotonics Inc., Hong Kong, China). This system uses the high-purity hydrogen and oxygen as the fuel to obtain a high heating temperature of up to 2500–3000 °C. When SMF reaches a melting state at high temperature, its two ends were fixed onto two motorized displacement platforms and stretched in opposite directions. By carefully adjusting the speed and scanning region of the flame, the microfiber with uniform diameter can be obtained in the heating zone. Different diameters were easily achieved by controlling the stretching velocity. Fabry-Perot interferometer was finally fabricated by assistance of a homemade micromanipulation system (inset (c) of Figure 1). A cut of transparent HCF was fixed on a slide glass substrate with UV glue. One end of SMF and microfiber were cut with a flat-face and acted as two reflecting surfaces of Fabry-Perot interferometer. The other tail-ends of SMF and microfiber were clamped by two fiber claps and fixed onto two three-dimensional (3-D) optical fiber adjusting frames (APFP-XYZ, adjusting precision <2 μm, Zolix Instruments Co., Ltd., Beijing, China).

In this case, the Fabry-Perot structure can be timely observed and measured by a microscope system (DMM-300C, Shanghai Caikon Optical Instrument Co., Ltd., Shanghai, China) on a computer and its cavity length was also timely precisely manipulated according to the reflected spectrum. The basic component and curing agent were mixed with a weight ratio of 10:1 to obtain the PDMS sol, which was filled into the HCF using a syringe (inset (d) of Figure 1) and cured in ~20 min. The experimental schematic was illustrated in Figure 2.

An amplified stimulated emission (ASE, ASE-C light source, 1520–1610 nm, Shenzhen Golight Technology Co., Ltd., Shenzhen, China) was used as the light source. The light was launched into the Fabry-Perot interferometer temperature probe through a 1 × 2 coupler (with the splitter ratio of 50:50). The reflection optical signal was collected by an optical spectrum analyzer (OSA, AQ6370, 600–1700 nm, resolution 20 pm, Yokogawa Electric Corp., Tokyo, Japan). In this work, the optical polarization direction does not affect the sensing performance due to the circularly polarized light output of ASE source and the cylindrical structure of microfiber. The temperature probe was placed in a thermostat (25–250 °C, resolution 0.1 °C, Shanghai Boxun Medical Biological Instruments Co., Ltd., Shanghai, China). The inset is a micrograph of the proposed temperature probe. The microfiber has a uniform diameter of ~63 μm and a length of ~2 cm. This length should be carefully controlled depending on the cone angle of the microfiber taper. Too long and thin microfiber will be easily adsorbed on the inner surface of HCF because of van der Waals force. In this case, it will be difficult to parallel its end-face with the reflecting surface of SMF to construct the two reflectors of Fabry-Perot interferometer. The cavity length was finally determined as ~34 μm.

## 3. Results

In the experiment, the Fabry-Perot interferometer temperature probe was placed in a thermostat. The temperature was increased from room temperature to 100 °C with steps of 1 °C. The reflection spectra of the Fabry-Perot temperature probe were recorded by a spectrometer. The printed pictures of the spectrometer screen at different temperature (40 °C and 41 °C) are illustrated in Figure 3. The spectrum curve refers to the original spectrum reflected from the microfiber. The free spectral range (FSR) is ~21 nm, during which the wavelength values of the resonance dips were determined with demodulation equipment with the resolution of 1 pm. When temperature changed from 40 °C (Figure 3a) to 41 °C (Figure 3b), the resonance dip moved with a wavelength location shift of ~10.5 nm, indicating a resolution of lower than 10^−4^ °C. Due to the limitation of one period of FSR, the ultra-sensitive temperature fluctuation monitoring can be achieved in the maximum range of 0 °C to ~2 °C (fluctuation level: ±1 °C). In order to achieve a commercial low-cost device, a photodiode can be used to monitor the change in intensity of a single wavelength to determine the direction and magnitude of temperature fluctuations.

The sensitivity of Fabry-Perot interferometer is dependent on the resonance shift as a function of temperature [38]:(1)$$s=\frac{\Delta \lambda_{T}}{\Delta T}=\lambda_{m}\alpha_{FP}=\lambda_{m}\left(\frac{L_{PDMS}\alpha_{PDMS,T}-L_{fiber}\alpha_{fiber,T}}{L_{PDMS}}\right)$$
where, *α**_FP_* refers to the relative change for the cavity length of the Fabry-Perot interferometer. In this work, it is depended on both the thermal expansion of PDMS and silica fibers. The FSR can be expressed as:(2)$$FSR=\frac{\lambda^{2}}{2n_{PDMS}L_{PDMS}}$$

In addition to the incident wavelength, FSR is inversely proportional to the change in refractive index and length of PDMS, which are determined by its thermo-optic coefficient and thermal expansion coefficient, respectively. Here, the thermal expansion coefficient plays a dominant role in the temperature change process, since the bulk expansion of PDMS is limited by the HCF wall and transferred into a change in cavity length to improve the sensitivity of the sensor. For the proposed Fabry-Perot interferometer, FSR is inversely proportional to the spacing between the microfiber and SMF tips, which was demonstrated experimentally when we continuously moved the microfiber towards the SMF in the HCF.

To demonstrate the temperature sensing performance in a wider range, the wavelength movement of one resonance dip was marked and traced in the whole spectrum range of the light source from 1520 nm to 1610 nm, as shown in Figure 4. When the temperature increased from 43 °C to 50 °C with steps of 1 °C, a resonance dip was marked to trace its shift amount. The inset of Figure 4 illustrates eight reflection spectra recorded at the different temperature values, where the resonance dip is red-shifted for almost a half cycle of the FSR. This resonance dip was independently selected to clearly display the temperature sensing characteristics from 43 °C to 50 °C. The resonance dip red-shifted continuously from 1534.8 nm (43 °C) to 1607.3 nm (50 °C).

The corresponding temperature sensing characteristic curve was shown in Figure 5. Black (circle) and red (pentagon) experimental data points and corresponding linear fitting represent the temperature response results for heating and cooling process, respectively.

During the heating process, a sensitivity of up to 10.37 nm/°C for the temperature sensing was experimentally demonstrated with a linearity of 0.99965. To verify the recovery characteristics of the temperature sensor, the movement of resonance dip was recorded through the cooling process in the same temperature range (from 50 °C to 43 °C in steps of 1 °C). By linearly fitting the experimental data points, a sensitivity of up to 10.67 nm/°C was obtained with a linearity of 0.99535. The performance curve illustrates the relationship between resonance dip and temperature, which will be stable for a temperature probe with fixed structure parameters. When a new sensor is used, the temperature can be determined by referring to calibration curve.

To reveal the repeatability and stability of the proposed temperature sensor, three-cycle experiments for a sensing probe with the cavity length of 31 μm and the microfiber diameter of 61 μm were performed, where the corresponding wavelength shift values depending on the temperature increasing/decreasing were recorded and illustrated in Figure 6. The highest sensitivity of 11.86 nm/°C was experimentally demonstrated for the temperature increase process in the first round, which was higher than the probe in Figure 5 mainly due to the shorter cavity, which matches well with the theoretical analysis. Equation (1) indicates that the sensitivity is proportional to the relative change in cavity length. A shorter cavity will result in a more significant change than that of the longer one. Furthermore, its larger FSR enables high-precise temperature fluctuation monitoring in a wider range (see the analysis of Equation (2) and Figure 3).

The maximum wavelength backlash was determined as ~1.3 nm during the three-cycle measurement process. On the one hand, this is related to the thermal expansion relaxation time of PDMS; on the other hand, it is also limited by the temperature control accuracy of the oven, which is also indicated in the stability measurement of the proposed temperature probe (inset of Figure 6). When the temperature fluctuates between 45 °C and 46 °C, the positional fluctuation of the resonance wavelength was less than ~0.2 nm, and the corresponding response time (stabilization time) was ~3 min. The above fluctuations fall within the performance range of the thermostatic oven. In order to calibrate the sensing characteristics of this temperature probe in a larger working range, the specific resonance dips should be dynamically selected in different temperature ranges. Thereafter, the temperature sensing characteristic curve can be obtained by using the relative shift of the labeled resonance dips. In addition to the microfiber and SMF, the final working range of this temperature probe will be limited by the sensitive materials. PDMS in solid status has the stable physicochemical property in the temperature range of −55–200 °C. Therefore, this temperature probe can work in a wider range, not limited to the results reported in this work.

## 4. Discussion

The optical fiber temperature sensor proposed in this work is compact and easily prepared. Its sensitivity is significantly higher than most of other fiber temperature sensors reported in recent years, as compared in Table 1.

As can be seen from Table 1, the temperature sensitivity of FBGs is low, and the encapsulation technology and demodulation optical path are complex [16]. The dual-arms system of Mach-Zehnder interferometers are commonly built using special optical fibers (for example PCF [6,20] or microfibers [18,19,30]) or by splicing different optical fibers [18,21], where the sensitive liquid or polymer were introduced to create a temperature-sensitive probe [6,20,21]. In contrast, the Fabry-Perot fiber interferometer can be easily fabricated on a single fiber. It has a more compact structure for developing high-performance temperature microprobes. Femtosecond laser [31] or ion beam etching technology [32], as well as high-precision fiber-splicing technology [36,37], can improve its temperature detection limit to as high as 1200 °C, making it suitable for extreme high temperature environments; furthermore, sol coating [35] or temperature-sensitive polymer encapsulation technology [39] can be used for enhance normal temperature microprobes, which will be a promising candidate for implantable microsensors for health or environmental monitoring under 200 °C.

Compared with the polymer film reflector, in this work, the smooth end-faces of SMF and microfiber were used as the two reflectors of the Fabry-Perot interferometer. PDMS is used to fix the two reflectors and realize a highly sensitive response to temperature changing. The temperature response properties can be revealed by the contribution of the negative thermal-optics coefficient (*α_to_*: −450 × 10^−6^/°C) and the thermal-expansion coefficient (*α_te_*: 960 × 10^−6^/°C) effects of PDMS. When the temperature increases, a smaller effective refractive index and a longer cavity length will be obtained, respectively. In view of their contributions to the effective optical path between the two reflectors of the Fabry-Perot interferometer, they have the opposite impact on the cavity length when the temperature changes.

## 5. Conclusions

In this paper, a compact and miniature Fabry-Perot interferometer based on a microfiber and SMF in a cut of HCF was proposed and experimentally demonstrated. The morphology parameters, such as microfiber diameter and cavity length, can be precisely controlled by the microfiber fabrication (scanning flame stretching technique) and micromanipulation processes (microscope- assised micromanipulation method), respectively. By filling PDMS into this Fabry-Perot interferometer with the microfiber diameter of ~63 μm and cavity length of ~34 μm, a temperature sensitivity of higher than 10 nm/°C was experimentally obtained. When the cavity length was reduced to ~31 μm, a highest sensitivity of 11.86 nm/°C has been experimentally demonstrated with an excellent repeatability and stability. Due to its high sensitivity and easily adjustable morphology, this Fabry-Perot temperature sensor has promising applications for precisely monitoring temperature fluctuations in biochemical reaction processes, industrial production and food storage.

## Figures and Tables

**Figure 1 sensors-19-01819-f001:**
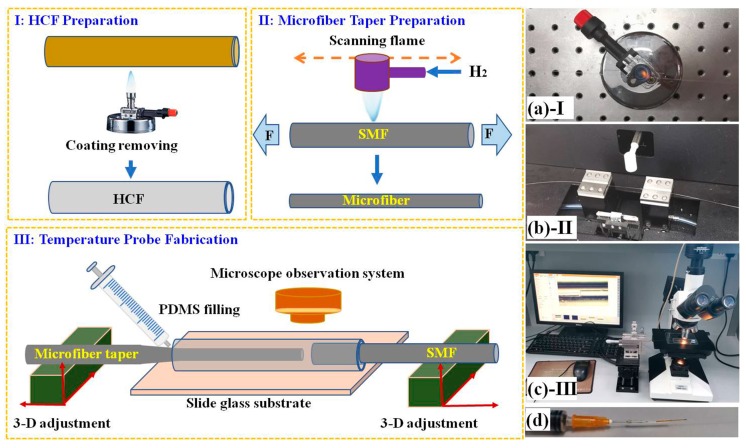
Fabrication process of the microfiber and PDMS based Fabry-Perot temperature probe. I: Coating layer of HCF was removed to prepare the transparent HCF (inset (**a**)); II: MF taper was prepared by scanning flame stretching technique (inset (**b**)); III: Fabry-Perot temperature probe was fabricated by assistance of the micromanipulation method under a microscope (insets (**c**) & (**d**)).

**Figure 2 sensors-19-01819-f002:**
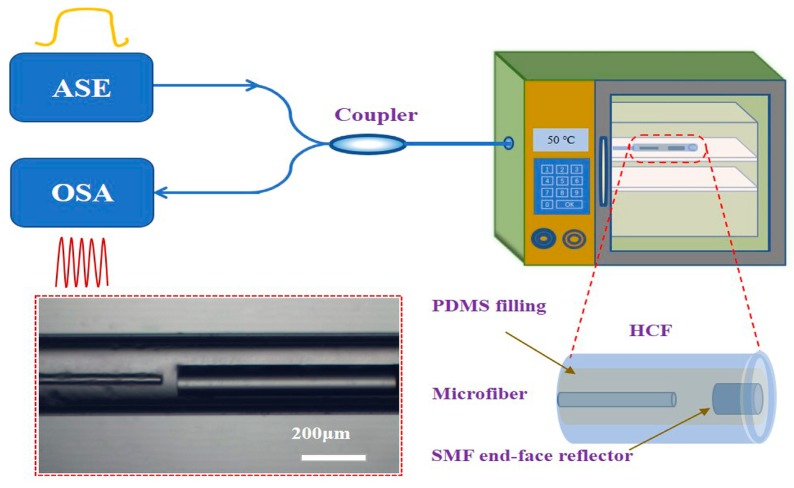
Experimental schematic of the microfiber and PDMS-based Fabry-Perot interferometer for sensing temperature. The light source, temperature sensor and spectrometer were contacted by a 1 × 2 coupler. The enlarged schematic and microscope picture of the temperature probe were illustrated.

**Figure 3 sensors-19-01819-f003:**
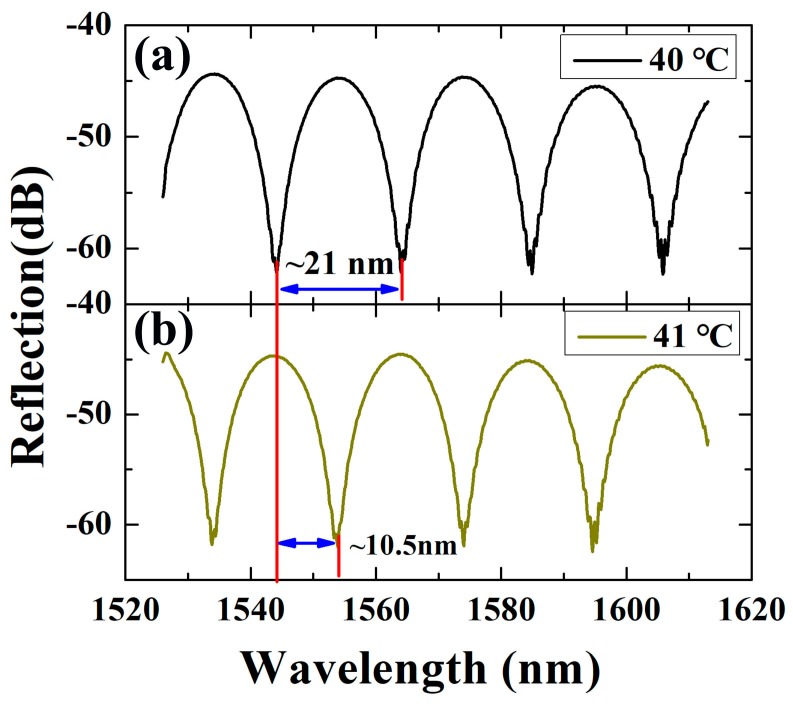
Reflection spectra of the microfiber Fabry-Perot temperature probe at temperature of (**a**) 40 °C and (**b**) 41 °C, respectively.

**Figure 4 sensors-19-01819-f004:**
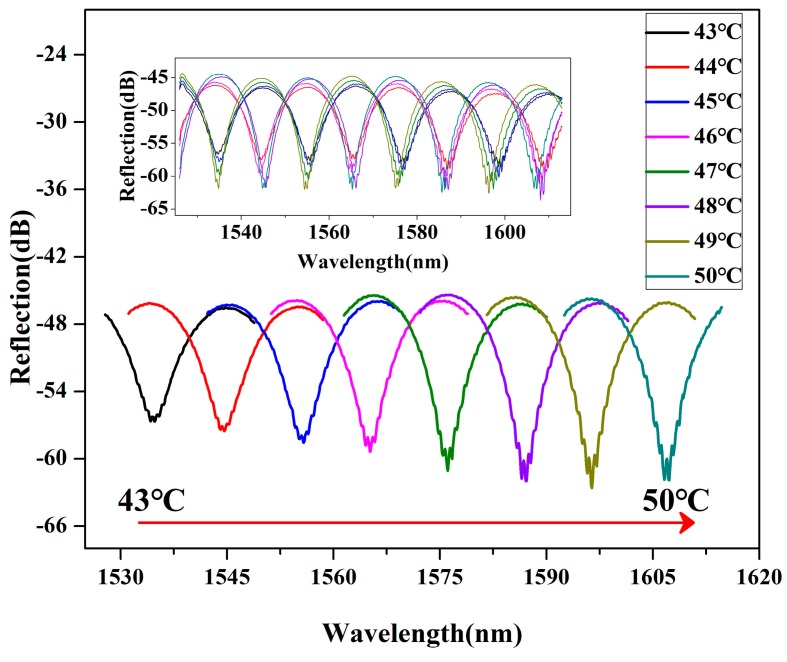
Reflection spectra of the microfiber Fabry-Perot temperature probe for the temperature increased from 43 °C to 50 °C with a step of 1 °C.

**Figure 5 sensors-19-01819-f005:**
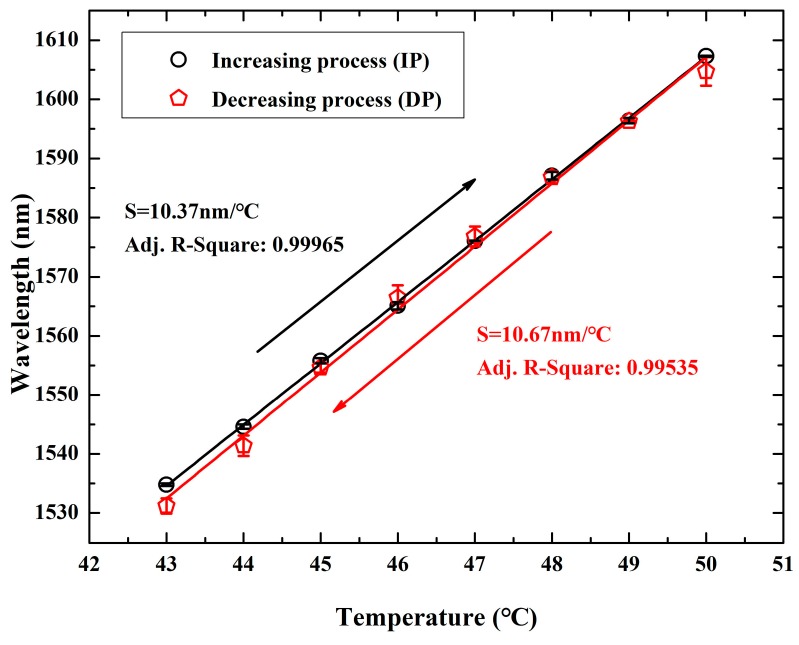
Location of resonance dip changed as a function of temperature with an excellent linear fitting during the increasing and decreasing process of 43–50 °C. The corresponding sensitivities were determined to be 10.37 nm/°C and 10.67 nm/°C, respectively.

**Figure 6 sensors-19-01819-f006:**
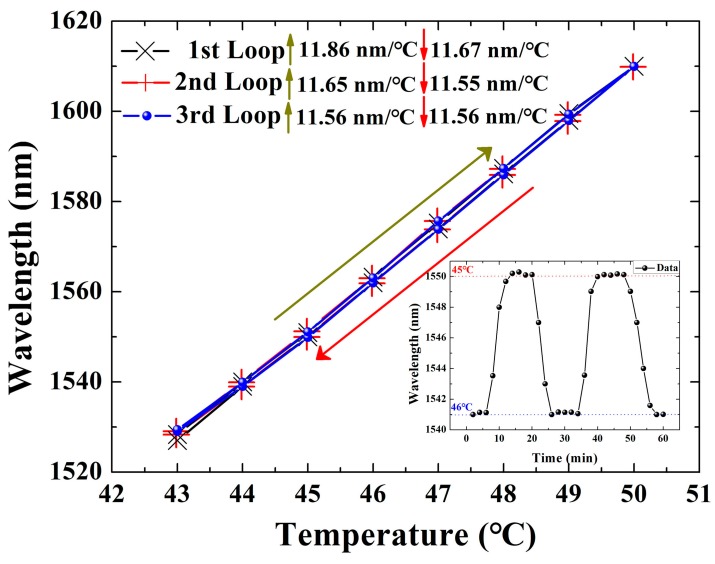
Three-cycle experiments for the temperature-dependence curves for a Fabry-Perot interferometer with the cavity length of 31 μm and the microfiber diameter of 61 μm. Inset: Stability of the wavelength locations for the temperature changing between 45 °C and 46 °C.

**Table 1 sensors-19-01819-t001:** Sensing performance comparison for typical temperature probes based on optical fibers.

Mechanism	Structure	Sensitivity	Range	Reference
Grating interference	Copper tube/FBG	27.6 pm/°C	0–35 °C	[16]
	FBG	18.8 pm/°C	20–90 °C	[17]
Mach-Zehnder interference	SMS/Microfiber	6.5 nm/°C	51–65 °C	[18]
	Micro-bend fiber	1.92 × 10^−^^3^/°C	29–52 °C	[19]
	SMS/Liquid	−1.88 nm/°C	0–80 °C	[21]
	Liquid cored PCF	−2.15 nm/°C	20–80 °C	[6]
	Liquid-filled PCF	−1.83 nm/°C	23–58 °C	[20]
	C-typed PCF	−7.609 nm/°C	15–30 °C	[23]
	NOA 73/PMMA	−431 pm/°C	25–75 °C	[22]
	PMMA	1.04 × 10^−3^/°C	25–120 °C	[27]
	Abrupt tapered fiber	0.0833 dBm/°C	30–50 °C	[30]
Fabry-Perot interference	Single RI turning dot	13.9 pm/°C 18.6 pm/°C	100–500 °C 500–1000 °C	[31]
	Open microcavity	−654 pm/°C	30–120 °C	[32]
	HC-PBF/HCF splicing	17 nm/°C	100–800 °C	[36]
	SMF/PCF splicing	15.61 pm/°C	300–1200 °C	[37]
	LOCTITE 3493 film	~5.2 nm/°C	15–22 °C	[35]
	Microfiber taper	1.97 pm/°C	50–150 °C	[38]
	Nafion film	2.71 nm/°C	−15–65 °C	[39]
	Microfiber/SMF/PDMS	10.67 nm/°C	43–50 °C	This work

SMS: Single-muti-single mode fiber; NOA 73: Norland optical adhesive 73; RI: Refractive index; PMMA: poly(methyl methacrylate; LOCTITE 3493: Light cure adhesive 3493.

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
