# Peer review of "A High Sensitivity Temperature Sensing Probe Based on Microfiber Fabry-Perot Interference"

_sensors, 2019, doi:10.3390/s19081819_

Reviewer 1 Report

This manuscript presented a Fabry-Perot sensing probe consist of the microfiber in a hollow core silica fiber (HCF) capillary for temperature measurement. My comments are as following:

1.     Although a literature survey was reported by the authors, there are many new optical fiber temperature sensing technique which have been reported recently. We also suggest the authors to give a brief summary of the literature survey to point out the missing part and its significance.

2.     The authors should point out the advantages of the proposed sensor system in comparison with conventional temperature sensors.

3.     As the paper is presented on the basis of experiments, I suggest the authors made a detailed analysis of several performance factors of the proposed sensing device. These are:

l   temperature dependence

l   Polarization dependence

l   Stability and repeatability

4.      The author should show the picture of the sensor manufacturing process.

5.      The author should explain the theoretical formula and discuss about the effect of the spacing between the microfiber taper and the SMF.

6.      In Figure 3, Please explain why the noise is very large at 1520 nm and 1620nm.

7.      Referring to the Figure 5, the calibration experiment seems to be done just one time. Three-time (3 cycles) experiments should be performed to show the reproducibility.

8.      In Fig. 4-5, the author needs to increase more discussion on the comparison between the experimental results and the theoretical results.

Author Response

Reviewer 1:

1. Although a literature survey was reported by the authors, there are many new optical fiber temperature sensing technique which have been reported recently. We also suggest the authors to give a brief summary of the literature survey to point out the missing part and its significance.

Reply: We have reviewed the typical fiber sensor techniques and highlighted their significances in the revised manuscript.

See Page 1, Introduction section, Paragraph 1, lines 34-43:

"By combining the resonance enhancement effect of the optical coupling technique, multi-modes interference, optical evanescent field, optical time domain reflecting and optical ring-down technology produced by different special optical fiber structures, various optical fiber temperature sensors were realized [6-9]. Muti-modes interference is carried out by splicing together different kinds of fibers to excite the modes interference. The splicing joints are fragile and the length for each section must be carefully controlled during the fabrication process; Optical evanescent field can be obtained around the micro/nanofibers with the compared diameters with that of incident light. Although the micro/nanofibers contribute the excellent performance, the sensor probes based on them were difficult to fabricate because of their thin diameter and environment sensitive properties; Optical time domain reflecting technique was used to sense temperature and strain based on the Raman or Brillion scattering, where the sensing performance improvement relies on using some special fibers [10]; The sensitivity of the temperature sensor based on optical ring-down technology only can be increased by extending the fiber length."

See Pages 1-2, Introduction section, Paragraph 2, lines 44-48:

"In addition to the basic sensing mechanism, their sensing performance was further improved by means of temperature sensitive materials [11]. Many materials, such as polymer and metal oxide, have been reported to elaborate by surface or inner coating, and encapsulate the whole fiber structures [12-14]. In addition to the temperature dependence, the effect of humidity, strain and other related parameters on the sensing performance must be revealed and eliminated. "

See Page 2, Introduction section, Paragraph 3, lines 60-69:

"However, the structures of the Mach-Zehnder interferometers are complex due to their dual-optical-paths system[19-22]. To simplify the structures, the two optical paths can be revealed in single fiber, named as in-line Mach-Zehnder interferometer, such as C-typed PCF [23], side-hole PCF [24], D-shaped-hole fiber [25] and muti-core fiber [26], these compact structures were precisely machined using femtosecond laser, focused ion beam and chemical vapor deposition, and performed a excellent stability and sensing performance. However, it is hard to manufacture in batches due to the high cost and technical requirments. In addition to above complex optical fiber structures, the single polymer optical fiber have been demonstrated with a temperature sensitivity of ~10-3 °C [27], where the temperature performance were revealed by the transmission power and the effect of relative and twist have been experimentally obtained [28, 29]. "

2. The authors should point out the advantages of the proposed sensor system in comparison with conventional temperature sensors.

Reply: The advantage of the proposed sensor has been discussed by comparing it with conventional temperature sensors. See Page 2, Introduction section, Paragraph 4

"Comparing with the conventional temperature sensor, the proposed Fabry-Perot interferometer temperature sensor cost lower and was easier and fast to prepare. This compact Fabry-Perot temperature probe was proposed by encapsulating microfiber and single mode fiber (SMF) tip in a hollow core fiber (HCF), between which the temperature sensitive poly-dimethylsiloxane (PDMS) was filled and cured. Microfiber was prepared by the one-step heating-stretching technique from the normal SMF. Microfiber and SMF can be easily lined due to the compared inner diameter of HCF. The high transparent and low refractive index of PDMS caused little impact on the incident light. Furthermore, a sensitivity of higher then 11 nm/°C has been experimentally demonstrated due to its high thermal expansion coefficient. This temperature sensor will be a promising candidate for monitoring the temperature fluctuation in some small space due to its high sensitivity and scale (200 μm in diameter and<5mm in length)."

3. As the paper is presented on the basis of experiments, I suggest the authors made a detailed analysis of several performance factors of the proposed sensing device. These are:

l   temperature dependence

l   Polarization dependence

l   Stability and repeatability

Reply: The above performance factors for the proposed sensor have been analyzed in the revised manuscript. The repeatability and stability have been experimentally performed in Fig. 6(a) and Fig. 6(b). See Page 4, lines 138-140:

"... ... In this work, the optical polarization direction does not affect the sensing performance due to the circularly polarized light output of ASE source and the cylindrical structure of microfiber.... ..."

See Page 5, lines 172-179:

"In addition to the incident wavelength, FSR is inversely proportional to the change in refractive index and length of PDMS, which are determined by thermo-optic coefficient and thermal expansion coefficient, respectively. Here, the thermal expansion coefficient plays a dominant role in the temperature change process, since the bulk expansion of PDMS is limited by the HCF wall and transferred into the change in cavity length to improve the sensitivity of the sensor. For the proposed Fabry-Perot interferometer, FSR is inversely proportional to the spacing between microfiber and SMF tips, which were demonstrated experimentally when we continuously moved the microfiber towards the SMF in the HCF."

See Fig. 6 and main text on Page 6, lines 208-229:

"To reveal the repeatability and stability of the proposed temperature sensor, the three-cycle experiments for a sensing probe with the cavity length of 31 μm and the microfiber diameter of 61 μm were performed, where the corresponding wavelength shift values depending on the temperature increasing/decreasing were recorded and illustrated in Fig. 6. The highest sensitivity of 11.86 nm/℃ was experimentally demonstrated for the temperature increasing process in the first round, which was higher than the probe in Fig. 5 mainly due to the shorter cavity, which meets well with the theoretical analysis. Where, Eq. 1 indicates that the sensitivity is proportional to the relative change in cavity length. Shorter cavity will result in a more significant than that of the longer one. Furthermore, its larger FSR enables the high-precise temperature fluctuation monitoring in a wider range (see the analysis of Eq. 2 and Fig. 3)."

"The maximum wavelength backlash was determined as ~1.3 nm during the three-cycle measurement process. On the one hand, this is related to the thermal expansion relaxation time of PDMS; on the other hand, it is also limited by the temperature control accuracy of the oven, which is also indicated in the stability measurement of the proposed temperature probe (Inset of Fig. 6). When the temperature fluctuates between 45 ℃ and 46 ℃, the positional fluctuation of the resonance wavelength was less than ~0.2 nm, and the corresponding response time (stabilization time) was ~3 min. The above fluctuations fall within the performance range of the thermostatic oven.... ..."

4. The author should show the picture of the sensor manufacturing process.

Reply: The manufacturing process have been included in Fig. 1 and numbered as Inset (a): HCF coating removing above Bunsen burner, (b): Microfiber preparing, (c): Homemade micromanipulation system and (d): PDMS filling process. See Figure 1, on Page 3:

Figure 1. Fabrication process of the microfiber and PDMS based Fabry-Perot temperature probe. I: Coating layer of HCF was removed to prepare the transparent HCF (Inset (a)); II: MF taper was prepared by scanning flame stretching technique (Inset (b)); III: Fabry-Perot temperature probe was fabricated by assistance of the micromanipulation method under a microscope (Inset (c) & (d)).

5. The author should explain the theoretical formula and discuss about the effect of the spacing between the microfiber taper and the SMF.

Reply: The theoretical formula has been included and discussed. The effect of the spacing on the interference spectra has been analyzed. See Pages 4-5, lines 162-179,

"The sensitivity of Fabry-Perot interferometer is dependent on the resonance shift as a function of temperature [31]

            (1)

where, αFP refers to the relative change for the cavity length of the Fabry-Perot interferometer. In this work, it is depended on both the thermal expansion of PDMS and silica fibers. The FSR can be expressed as

                                   (2)

In addition to the incident wavelength, FSR is inversely proportional to the change in refractive index and length of PDMS, which are determined by thermo-optic coefficient and thermal expansion coefficient, respectively. Here, the thermal expansion coefficient plays a dominant role in the temperature change process, since the bulk expansion of PDMS is limited by the HCF wall and transferred into the change in cavity length. For the proposed Fabry-Perot interferometer, FSR is inversely proportional to the spacing between microfiber and SMF tips, which were demonstrated experimentally when we continuously moved the microfiber towards the SMF in the HCF."

6. In Figure 3, Please explain why the noise is very large at 1520 nm and 1620nm.

 Reply: The output wavelength of the light source used in this work is 1520-1610. That is why the noise is large at its both edges. We have revised Figure 3 to eliminate the confuse information. See Figure 3 on Page 5:

Figure 3. Reflection spectra of the microfiber Fabry-Perot temperature probe at temperature of (a) 40 °C and (b) 41 °C, respectively.

7. Referring to the Figure 5, the calibration experiment seems to be done just one time. Three-time (3 cycles) experiments should be performed to show the reproducibility.

 Reply: Another temperature sensor was prepared and calibrated by 3 cycles' experiments. The corresponding experimental results have been indicated as Fig. 6(a). See Fig. 6 (a), lines 208-229:

Figure 6. Three-cycle experiments for the temperature-dependence curves for a Fabry-Perot interferometer with the cavity length of 31 μm and the microfiber diameter of 61 μm. Inset: Stability of the wavelength locations for the temperature changing between 45 ℃ and 46 ℃.

"To reveal the repeatability and stability of the proposed temperature sensor, the three-cycle experiments for a sensing probe with the cavity length of ~31 μm and the microfiber diameter of 61 μm were performed, where the corresponding wavelength shift values depending on the temperature increasing/decreasing were recorded and illustrated in Fig. 6. The highest sensitivity of 11.86 nm/℃ was experimentally demonstrated for the temperature increasing process in the first round, which was higher than the probe in Fig. 5 mainly due to the shorter cavity, which meets well with the theoretical analysis. Where, Eq. 1 indicates that the sensitivity is proportional to the relative change in cavity length. Shorter cavity will result in a more significant than that of the longer one. Furthermore, its larger FSR enables the high-precise temperature fluctuation monitoring in a wider range (see the analysis of Eq. 2 and Fig. 3)."

"The maximum wavelength backlash was determined as ~1.3 nm during the three-cycle measurement process. ... ... "

8. In Fig. 4-5, the author needs to increase more discussion on the comparison between the experimental results and the theoretical results.

Reply: We have discussed and compared the experimental results with the theoretical results in the revised manuscript. See Page 5, lines 172-179, "In addition to the incident wavelength, FSR is inversely proportional to the change in refractive index and length of PDMS, which are determined by thermo-optic coefficient and thermal expansion coefficient, respectively. Here, the thermal expansion coefficient plays a dominant role in the temperature change process, since the bulk expansion of PDMS is limited by the HCF wall and transferred into the change in cavity length to improve the sensitivity of the sensor. For the proposed Fabry-Perot interferometer, FSR is inversely proportional to the spacing between microfiber and SMF tips, which were demonstrated experimentally when we continuously moved the microfiber towards the SMF in the HCF."

See Page 6, lines 211-217, "The highest sensitivity of 11.86 nm/℃ was experimentally demonstrated for the temperature increasing process in the first round, which was higher than the probe in Fig. 5 mainly due to the shorter cavity, which meet well with the theoretical analysis. Where, Eq. 1 indicates that the sensitivity is proportional to the relative change in cavity length. Shorter cavity will result in a more significant than that of the longer one. Furthermore, its larger FSR enables the high-precise temperature fluctuation monitoring in a wider range (see the analysis of Eq. 2 and Fig. 3)."

Reviewer 2 Report

Thanks for inviting me to review this paper. The paper proposes a new FPI temperature sensor. The sensor achieves a high temperature sensitivity due to the thermal expansion of PDMS. The paper is well written. Here are some comments for further improvement:

(1) In the introduction section, please explain the novelty more clearly. In the past several years, there are many studies on FPI temperature sensors. Please clarify your contribution. More references should be added. Here is a recent review paper for example:

Bao, Y.; Huang, Y.; Hoehler, M.; Chen, G. (2019). "Review of fiber optic sensors for structural fire engineering", Sensors, 19(4), 877; https://doi.org/10.3390/s19040877

(2) The new sensor is interesting in terms of the sensitivity and the linear relationship. However, the investigated temperature range (Figure 5) is form 43 to 50 degree C, which is too small and limits the significance of the sensor.

(3) Why did you use hollow core silica fiber?

(4) How did you determine the temperature from the measured curves from the new sensor? What are the theories that support your method?

(5) Please keep the formats consistent - Figure 3 has a different format from the other figures (4 and 5). 

Author Response

Reviewer 2:

Thanks for inviting me to review this paper. The paper proposes a new FPI temperature sensor. The sensor achieves a high temperature sensitivity due to the thermal expansion of PDMS. The paper is well written. Here are some comments for further improvement:

(1) In the introduction section, please explain the novelty more clearly. In the past several years, there are many studies on FPI temperature sensors. Please clarify your contribution. More references should be added. Here is a recent review paper for example:

Bao, Y.; Huang, Y.; Hoehler, M.; Chen, G. (2019). "Review of fiber optic sensors for structural fire engineering", Sensors, 19(4), 877; https://doi.org/10.3390/s19040877

Reply: More works has been reviewed in the introduction section. The significance of the proposed temperature probe has also been explained.

See Page 1, lines 34-43:

"By combining the resonance enhancement effect of the optical coupling technique, multi-modes interference, optical evanescent field, optical time domain reflecting and optical ring-down technology produced by different special optical fiber structures, various optical fiber temperature sensors were realized [6-9]. Muti-modes interference is carried out by splicing together different kinds of fibers to excite the modes interference. The splicing joints are fragile and the length for each section must be carefully controlled during the fabrication process; Optical evanescent field can be obtained around the micro/nanofibers with the compared diameters with that of incident light. Although the micro/nanofibers contribute the excellent performance, the sensor probes based on them were difficult to fabricate because of their thin diameter and environment sensitive properties; Optical time domain reflecting technique was used to sense temperature and strain based on the Raman or Brillion scattering, where the sensing performance improvement relies on using some special fibers [10]; The sensitivity of the temperature sensor based on optical ring-down technology only can be increased by extending the fiber length."

See Pages 1-2, lines 44-48:

"In addition to the basic sensing mechanism, their sensing performance was further improved by means of temperature sensitive materials [11]. Many materials, such as polymer and metal oxide, have been reported to elaborate by surface or inner coating, and encapsulate the whole fiber structures [12-14]. In addition to the temperature dependence, the effect of humidity, strain and other related parameters on the sensing performance must be revealed and eliminated. "

See Page 2, lines 60-69:

"However, the structures of the Mach-Zehnder interferometers are complex due to their dual-optical-paths system [19-22]. To simplify the structures, the two optical paths can be revealed in single fiber, named as in-line Mach-Zehnder interferometer, such as C-typed PCF [23], side-hole PCF [24], D-shaped-hole fiber [25] and muti-core fiber [26], these compact structures were precisely machined using femtosecond laser, focused ion beam and chemical vapor deposition, and performed a excellent stability and sensing performance. However, it is hard to manufacture in batches due to the high cost and technical requirments. In addition to above complex optical fiber structures, the single polymer optical fiber have been demonstrated with a temperature sensitivity of ~10-3 °C [27], where the temperature performance were revealed by the transmission power and the effect of relative and twist have been experimentally obtained [28, 29]. "

(2) The new sensor is interesting in terms of the sensitivity and the linear relationship. However, the investigated temperature range (Figure 5) is form 43 to 50 degree C, which is too small and limits the significance of the sensor.

Reply: In this work, the wavelength range of the light source is 1520 nm to 1610 nm, having a wavelength width of less than 90 nm. While the sensitivity for the proposed temperature probe is higher than 10 nm/℃, we must trace a resonance dip to calibrate the sensing performance curve. Therefore, the working range was limited less than 8 ℃. This working range can be further expanded by dynamically tracing the different resonance dips in the spectra. We have clarified this problem in the manuscript. See Page 7, lines 229-232,

"... ...In order to calibrate the sensing characteristics of this temperature probe in a larger working range, the specific resonance dips should be dynamically selected in different temperature ranges. Thereafter, the temperature sensing characteristic curve can be obtained by using the relative shift of the labeled resonance dips. ... ..."

(3) Why did you use hollow core silica fiber?

Reply: In this work, the PDMS was used as the temperature sensitive material and inserted between the microfiber and SMF. During the preparation of Fabry-Perot interferometer, the hollow core silica fiber was used to fill the liquid state PDMS, and fix the microfiber and SMF in-line. Furthermore, the restriction of the HCF wall can convert the bulk expansion of PDMS into radial expansion to improve the sensitivity of the sensor. We have explained the reasons in the manuscript. See Page 2, lines 87-92:

"... ... This compact Fabry-Perot temperature probe was proposed by encapsulating microfiber and single mode fiber (SMF) tip in a hollow core fiber (HCF), between which the temperature sensitive poly-dimethylsiloxane (PDMS) was filled and cured. Microfiber was prepared by the one-step heating-stretching technique from the normal SMF. Microfiber and SMF can be easily lined due to the compared inner diameter of HCF. ... ..."

See Page 5, lines 174-177:

"... ...Here, the thermal expansion coefficient plays a dominant role in the temperature change process, since the bulk expansion of PDMS is limited by the HCF wall and transferred into the change in cavity length to improve the sensitivity of the sensor.... ..."

(4) How did you determine the temperature from the measured curves from the new sensor? What are the theories that support your method?

Reply: After calibration process (Fig. 5 and Fig .6), the temperature sensing performance curve was obtained when the temperature values were measured by the thermostatic chamber. The performance curve illustrates the relationship between resonance dip and temperature, which will be stable for a temperature probe with fixed structure parameters. When the new sensor was used, the temperature can be determined referring to calibration curve. We have clarified this issue in the manuscript. See Pages 6-7, lines 204-207,

"... ... The performance curve illustrates the relationship between resonance dip and temperature, which will be stable for a temperature probe with fixed structure parameters. When the new sensor was used, the temperature can be determined referring to calibration curve."

(5) Please keep the formats consistent - Figure 3 has a different format from the other figures (4 and 5). 

Reply: We have changed the format of Figure 3 by using the same axis with those of Fig. 4 and Fig. 5. See Fig. 3 on Page 5:                                            

Figure 3. Reflection spectra of the microfiber Fabry-Perot temperature probe at temperature of (a) 40 °C and (b) 41 °C, respectively.

Reviewer 3 Report

The authors presented a miniature Fabry-Perot temperature probe by using PDMS to encapsulate the microfiber in a cut of HCF. Some revisions should be made as follows:

The introduction must be improved considering polymer optical fiber Technology, including works of refractive index gratings, intensity based, etc. There are many papers from different groups, like Arnaldo Leal Junior, and others.

Related with point 1, the table 1 should be completed with that.

The authors said that achieved a low cost sensor, however additional words must be added to the manuscript to justify how we can get a interrogations system to collect the data from the sensor and to be acceptable this sensor to be commercial.

A double checking in the English should be considered.

Author Response

Reviewer 3:

The authors presented a miniature Fabry-Perot temperature probe by using PDMS to encapsulate the microfiber in a cut of HCF. Some revisions should be made as follows:

1. The introduction must be improved considering polymer optical fiber Technology, including works of refractive index gratings, intensity based, etc. There are many papers from different groups, like Arnaldo Leal Junior, and others.

Reply: We have reviewed the typical fiber sensor techniques and highlighted their significances, including the polymer optical fiber temperature sensors in the revised manuscript.

See Page 1, lines 34-43:

"By combining the resonance enhancement effect of the optical coupling technique, multi-modes interference, optical evanescent field, optical time domain reflecting and optical ring-down technology produced by different special optical fiber structures, various optical fiber temperature sensors were realized [6-9]. Muti-modes interference is carried out by splicing together different kinds of fibers to excite the modes interference. The splicing joints are fragile and the length for each section must be carefully controlled during the fabrication process; Optical evanescent field can be obtained around the micro/nanofibers with the compared diameters with that of incident light. Although the micro/nanofibers contribute the excellent performance, the sensor probes based on them were difficult to fabricate because of their thin diameter and environment sensitive properties; Optical time domain reflecting technique was used to sense temperature and strain based on the Raman or Brillion scattering, where the sensing performance improvement relies on using some special fibers [10]; The sensitivity of the temperature sensor based on optical ring-down technology only can be increased by extending the fiber length."

See Pages 1-2, lines 44-48:

"In addition to the basic sensing mechanism, their sensing performance was further improved by means of temperature sensitive materials [11]. Many materials, such as polymer and metal oxide, have been reported to elaborate by surface or inner coating, and encapsulate the whole fiber structures [12-14]. In addition to the temperature dependence, the effect of humidity, strain and other related parameters on the sensing performance must be revealed and eliminated. "

See Page 2, lines 60-69:

"However, the structures of the Mach-Zehnder interferometers are complex due to their dual-optical-paths system [19-22]. To simplify the structures, the two optical paths can be revealed in single fiber, named as in-line Mach-Zehnder interferometer, such as C-typed PCF [23], side-hole PCF [24], D-shaped-hole fiber [25] and muti-core fiber [26], these compact structures were precisely machined using femtosecond laser, focused ion beam and chemical vapor deposition, and performed a excellent stability and sensing performance. However, it is hard to manufacture in batches due to the high cost and technical requirments. In addition to above complex optical fiber structures, the single polymer optical fiber have been demonstrated with a temperature sensitivity of ~10-3 °C [27], where the temperature performance were revealed by the transmission power and the effect of relative and twist have been experimentally obtained [28, 29]. "

2. Related with point 1, the table 1 should be completed with that.

Reply: The related results have been included in Table 1 as well. See Table 1, on Page 8, Ref. [27]

Table 1. Sensing performance comparison for typical temperature probes based on optical fibers

Mechanism

Structure

Sensitivity

Range

Reference

Grating   interference

Copper   tube/FBG

27.6   pm/°C

0-35   °C

[16]

FBG

18.8   pm/°C

20-90   °C

[17]

Mach-Zehnder   interference

SMS/Microfiber

6.5   nm/°C

51-65   °C

[18]

Micro-bend   fiber

1.92×103/°C

29-52   °C

[19]

SMS/Liquid

−1.88   nm/°C

0-80   °C

[21]

Liquid   cored PCF

−2.15   nm/°C

20-80   °C

[6]

Liquid-filled   PCF

−1.83   nm/°C

23-58   °C

[20]

C-typed PCF

-7.609 nm/°C

15-30 °C

[23]

NOA   73/PMMA

−431   pm/°C

25-75   °C

[22]

PMMA

1.04×103/°C

25-120 °C

[27]

Abrupt   tapered fiber

0.0833   dBm/°C

30-50   °C

[30]

Fabry-Perot   interference

Single   RI turning dot

13.9   pm/°C

18.6   pm/°C

100-500   °C

500-1000   °C

[31]

Open   microcavity

−654 pm/°C

30-120   °C

[32]

HC-PBF/HCF splicing

17 nm/°C

100-800   °C

[36]

SMF/PCF   splicing

15.61   pm/°C

300-1200   °C

[37]

LOCTITE   3493 film

~5.2   nm/℃

15-22   °C

[35]

Microfiber taper

1.97 pm/°C

50-150 °C

[38]

Nafion   film

2.71   nm/°C

−15-65   °C

[39]

Microfiber/SMF/PDMS

10.67   nm/°C

43-50   °C

This work

3. The authors said that achieved a low cost sensor, however additional words must be added to the manuscript to justify how we can get a interrogations system to collect the data from the sensor and to be acceptable this sensor to be commercial.

Reply: We have clarified this issue in the revised manuscript. See Page 5, lines 157-161:  

"... ... Due to the limitation of one period of FSR, the ultra-sensitive temperature fluctuation monitoring can be achieved in the maximum range of 0 °C to ~2 °C (fluctuation level: ±1 °C). In order to achieve a commercial low-cost device, a photodiode can be used to monitor the change in intensity of a single wavelength to determine the direction and magnitude of temperature fluctuations."

4. A double checking in the English should be considered.

Reply: We have double checked the grammar and typos problems, fixed and marked them in blue color in the revised manuscript.

Round  2

Reviewer 1 Report

The authors have addressed the review comments. Hence, this manuscript is suitable for publication.

Author Response

Thank you for the recommendation of reviewer.

Reviewer 2 Report

Thanks for your reply.  My questions and comments have been well address.  However, in the revised manuscript, some statements are wrong:

Lines 39-42: "Optical time domain reflecting technique was used to sense temperature and strain based on the Raman or Brillion scattering, where the sensing performance improvement relies on using some special fibers [10]."

Although specialty fibers were used in some studies, telecommunication-grade single-mode optical fibers have been used in more cases.  Different research groups have reported consistent results and agree that normal fused silica single-mode fibers can be used.  In fact, the normal fibers are preferred.  

Here are two recent papers from two different groups in different countries for your reference:

Bao, Y. and Chen, G., 2016. High-temperature measurement with Brillouin optical time domain analysis of an annealed fused-silica single-mode fiber. Optics letters, 41(14), pp.3177-3180.

Delepine-Lesoille, S.; Planes, I.; Landolt, M.; Hermand, G.; Perrochon, O. Compared performances of Rayleigh, Raman, and Brillouin distributed temperature measurements during concrete container fire test. In Proceedings of the 25th International Conference on Optical Fiber Sensors, Jeju, Korea, 24–28 April 2017; Volume 10323, p. 103236.

Author Response

Thank you for the reviewer's attentions. We have fixed this problem by deleting the related text.

See Introduction section, Page 1, Lines 39-41: "... ... Optical time domain reflecting technique was used to sense temperature and strain based on the Raman or Brillion scattering [10]; ... ..."

Reviewer 3 Report

I am happy with the improvements.

Author Response

(The authors gave the same response as above.)
